# Mechanical Behavior of a Medium-Entropy Fe_65_(CoNi)_25_Cr_9.5_C_0.5_ Alloy Produced by Selective Laser Melting

**DOI:** 10.3390/ma16083193

**Published:** 2023-04-18

**Authors:** Elizaveta Povolyaeva, Dmitry Shaysultanov, Ilya Astakhov, Stanislav Evlashin, Margarita Klimova, Nikita Stepanov, Sergey Zherebtsov

**Affiliations:** 1Laboratory of Bulk Nanostructured Materials, Belgorod National Research University, 85 Pobeda Str., 308015 Belgorod, Russia; 2Skolkovo Innovation Center, 5 Str. Nobel, 121205 Moscow, Russia; 3World-Class Research Center “Advanced Digital Technologies”, State Marine Technical University, 198095 Saint Petersburg, Russia

**Keywords:** medium-entropy alloy, selective laser melting, TRIP effect, martensite

## Abstract

Specimens of a medium-entropy Fe_65_(CoNi)_25_Cr_9.5_C_0.5_ (in at.%) alloy were produced using additive manufacturing (selective laser melting, SLM). The selected parameters of SLM resulted in a very high density in the specimens with a residual porosity of less than 0.5%. The structure and mechanical behavior of the alloy were studied under tension at room and cryogenic temperatures. The microstructure of the alloy produced by SLM comprised an elongated substructure, inside which cells with a size of ~300 nm were observed. The as-produced alloy demonstrated high yield strength and ultimate tensile strength (YS = 680 MPa; UTS = 1800 MPa) along with good ductility (tensile elongation = 26%) at a cryogenic temperature (77 K) that was associated with the development of transformation-induced plasticity (TRIP) effect. At room temperature, the TRIP effect was less pronounced. Consequently, the alloy demonstrated lower strain hardening and a YS/UTS of 560/640 MPa. The deformation mechanisms of the alloy are discussed.

## 1. Introduction

Some high- and medium-entropy alloys (HEAs and MEAs) with a face-centered cubic (fcc) structure have a very good combination of strength and ductility at low temperatures. Consequently, these materials can be considered as potential candidates for application under cryogenic conditions [1,2,3]. At the same time, it is known that such alloys often have low yield stresses. To overcome this problem, HEAs and MEAs can be alloyed with interstitial elements such as C, N, etc. due to which both yield strength and strength-ductility ratio were reported to improve considerably [4,5,6,7,8,9]. 

Besides, the outstanding properties of some HEAs and MEAs at room and cryogenic temperatures can be associated with the activation of twinning-induced plasticity (TWIP) and/or transformation-induced plasticity (TRIP) effects. For example, Bae et al. [10] reported a cryogenic ultimate tensile strength of ~1.5 GPa and ductility of ~87% due to the strain-induced fcc to body-centered cubic (bcc) phase transformation in a Fe_60_Co_15_Ni_15_Cr_10_ MEA. H. Huang et al. [11] used the “metastability” strategy to produce a composite-like structure with alternating hexagonal close-packed (hcp) and mechanically unstable bcc phases in the Ta_x_HfZrTi HEAs. The phase transformation resulted in an intense strain hardening effect due to the dynamic distribution of strain and stress between the bcc and hcp phases and promoted plastic deformation within the grains, which effectively suppressed early cracking and resulted in an outstanding combination of strength and ductility. In an Fe_65_(CoNi)_25_Cr_9.5_C_0.5_ alloy, the development of the TRIP effect at cryogenic temperatures resulted in very high strength (~2.1 GPa) and good ductility of ~26% [7]. 

Unfortunately, many HEAs and MEAs often contain expensive components, which is one of the reasons why they are still far from industrial applications despite their excellent mechanical properties [10,11,12,13,14,15,16,17,18]. The most obvious way to reduce the cost of alloys is to decrease the fraction of more expensive elements (Ni, Co, etc.) by increasing the concentration of less expensive species (e.g., Fe). A similar approach has been widely applied for a variety of HEAs, and some of the proposed alloys demonstrated quite attractive properties [19,20]. For example, by adjusting the ratio of elements R. Wei et al. [12] obtained in an inexpensive Fe-Mn-Ni-Cr system a combination of tensile strength of ~0.98 GPa and ductility of ~83% at 77 K due to the joint contributions of the TRIP and TWIP effects.

In the last decades, additive manufacturing technologies have been increasingly used to produce metallic components with complex geometric shapes. One of the attractive additive manufacturing methods is selective laser melting (SLM), in which metal parts can be sintered directly based on a 3D model, regardless the of shape complexity, and sometimes without the need for tooling specific parts. Excellent characteristics can be achieved using SLM technology in various metallic materials, including aluminum alloys [21,22], martensitic steels [23], and HEAs [24,25,26,27,28,29,30,31,32,33]. More recently, it has been shown that the SLM method has the potential to produce HEAs and MEAs with the TRIP/TWIP effect, which would have properties superior to counterparts obtained by other more conventional methods. Most of these studies were aimed at achieving a good combination of strength and ductility using TRIP and TWIP effects in SLM-derived alloys [34,35,36,37,38,39,40].

Although significant progress has been made in the creation of HEA/MEAs using the SLM method, limited research has been conducted on HEA/MEA with the TRIP/TWIP effect. Most HEAs and MEAs formed by the SLM method do not have sufficiently good combinations of strength and ductility, and many features of the structure and properties of such alloys remain unexplored. An impressive ability to control the microstructure and the combination of strength and ductility prompted us to explore the potential of a medium-entropy alloy produced by SLM. Therefore, the aim of this investigation was to study the possibility of manufacturing a medium-entropy TRIP alloy of the Fe-Co-Ni-Cr-C system using the SLM method, with a reasonable balance of strength and ductility at low temperatures.

## 2. Experimental

The powder for SLM was obtained by atomization of the vacuum induction melted Fe_65_(CoNi)_25_Cr_9.5_C_0.5_ alloy using a US35 atomizer from ATO LAB. The rods (Ø10 mm) were placed in a chamber that was purged with inert gas (argon). The obtained particles of the alloy had a spherical shape with an average size of 35 μm (Figure 1a). The EDS (Energy-dispersive X-ray spectroscopy detector installed at a FEI Quanta 600 scanning electron microscope (SEM)) maps suggest a uniform distribution of the elements within the particles.

SLM of the alloy was carried out in an argon atmosphere using a 3D printer DMP Flex 200 (3D Systems Inc., Rock Hill, SC, USA); a 316L steel plate was used as a substrate. The flow rate of argon gas used as shielding gas was 5 L per minute. To improve the quality of the workpieces, the direction of laser movement was changed by 67° during fusion of each subsequent layer [41] (Figure 1b); it should be noted that each laser beam track was declined 67° from the macroscopic scanning direction (SD). The distance between tracks and the beam diameter were 50 μm and 70 μm, respectively. 

The measured composition of the alloy (determined using optical emission spectroscopy or energy dispersive spectrometry (EDS); the percentages of C, O, and N were determined by LECO thermal combustion analysis) is shown in Table 1.

The microstructure of the sintered samples was studied using an optical microscope (OM, Olympus GX71, Tokyo, Japan), a scanning electron microscope (SEM, Quanta 600 FEG, Prague, Czech Republic) equipped with an electron backscatter diffraction (EBSD) detector, and a transmission electron microscope (TEM, JEOL JEM-2100, Tokyo, Japan). The TEM samples were first mechanically thinned to 100 μm and then subjected to conventional two-jet electropolishing in a mixture of 90% CH_3_COOH and 10% HClO_4_ at 30 V at room temperature. X-ray diffraction (XRD) analysis was performed using an ARL-X’tra Thermo Fisher Scientific (Thermo Fisher Scientific, Portland, OR, USA) diffractometer with Cu-Kα radiation.

The specimens measured 100 × 25 × 25 mm^3^ for tensile tests were also produced using SLM. Tensile dog-bone-like flat specimens with a gage size of 16 × 3 × 1.5 mm^3^ were cut parallel to the TD (transverse direction) /SD plane (Figure 2). Tensile tests were performed at an initial strain rate of 10^−3^ s^–1^ at room and cryogenic temperatures using an Instron 5882 universal testing machine. At least three specimens were tested for each condition. The fracture toughness tests were performed at room and cryogenic temperatures using Charpy V-notch specimens measuring 55 × 8 × 2 mm^3^ on an Instron IMP460 pendulum impact machine. At least three samples were tested for each condition.

## 3. Results

### 3.1. Selection of the SLM Parameters

More than 50 attempts were made to establish the most appropriate parameters for the SLM of the Fe_65_(CoNi)_25_Cr_9.5_C_0.5_ alloy. In the first stage, monolayers of the alloy (measured ~10 × 10 mm^2^) were obtained using a laser beam power in the range of 100–300 W and laser scan speed in the range of 400–2200 mm per second (Figure 3). It was found that at high power values (>300 W) and low speeds (<400 mm/s), the SLM process was associated with the spattering of melted alloy and excessively deep melting of the substrate. In contrast, high speeds (>2000 mm/s) and low power (<150 W) resulted in a lack of adhesion with the substrate and increased porosity of the layers, thereby suggesting incomplete melting of the powder. Similar dependencies were previously reported in [42,43,44]. Based on the obtained results, seven modes were selected (outlined in red in Figure 3 and listed in Table 2).

In the second stage, specimens of the alloy measuring 10 × 10 × 10 mm^3^ were produced using the selected parameters. Visual inspection of the specimens did not reveal any cracks or any other flaws on the surfaces. Microstructure analysis using SEM suggested a reversed bell-shaped dependence of the volume fraction of pores on the energy density (Figure 4). The latter shows how much energy is necessary to melt a certain volume of metal [22,45,46] and can be calculated as
(1)E=P / (V·h·t),
where *P* is the laser beam power (W), *V* is the beam velocity (mm/s), *h* is the distance between the tracks (mm), and *t* is the layer thickness (mm). For the Fe_65_(CoNi)_25_Cr_9.5_C_0.5_ alloy, the energy density varied in a wide range from 20 to 444 J/mm^3^; however, the less porous specimens were produced in the interval of *E* = 75–97 J/mm^3^ (volume fraction of pores less than 0.5%). The minimum volume fraction of pores of 0.1% (this amount is quite comparable with that in the as-cast specimens as per data from our previous work [7]) was found in the specimens produced at *E* = 83 J/mm^3^ (*P* = 200 W and *V* = 1600 mm/s) (Figure 4). It should be noted, however, that specimen #4 produced at the same energy has noticeably higher porosity in comparison with specimen #7, thereby suggesting some dependence on the power/beam scan speed combination as well. 

### 3.2. Structure and Properties of Fe_65_(CoNi)_25_Cr_9.5_C_0.5_ Alloy Specimens Produced by SLM

The results of the X-ray diffraction analysis (XRD) suggested that the sample consisted of only the fcc phase; the presence of the bcc phase was not observed (Figure 5). The comparison with the XRD profile of the initial powder did not suggest any noticeable effect of SLM on the phase composition of the alloy.

A typical image of the as-produced specimen is shown in Figure 6a. Good quality of the surfaces can be noted. At a macroscopic scale, laser beam tracks declined 67° from the macroscopic scanning direction SD and ranged from 80 μm to 100 μm in width were observed after SLM (marked by black dotted lines in Figure 6b): in the center of the tracks, large irregular-shaped grains, and at the edges, smaller equiaxed or elongated shapes were formed. No cracks or pores were observed on the surfaces of the obtained specimens.

EBSD inverse pole figures (IPF) and phase maps of the program alloy obtained by SLM are shown in Figure 7a,b, respectively, in the form of quasi-3D images. The structure was obtained from three orthogonal faces of the sample perpendicular to the SD, building direction (BD), and TD. In the as-produced condition, the alloy consisted mainly of the fcc phase with a small fraction (~1%) of the bcc phase. Note that this amount is below the typical detection limit of the XRD method. The average grain size at each face was ≈15 μm, with a scatter of the smallest and largest grain sizes ranging from ≈1 μm to ≈170, respectively. The grains often had a complex irregular shape and a well-developed internal substructure. The (111) pole figure did not suggest the formation of a pronounced texture in the alloy produced by SLM (Figure 7c).

TEM analysis revealed that the microstructure of the alloy comprised an elongated substructure (in the TD direction), inside which cells with a size of ~300 nm can be observed (Figure 8). The formation of an elongated structure in metallic materials obtained by additive technologies is quite typical [47,48]. The increased dislocation density and cellular structure formation may be caused by residual stresses arising from rapid solidification [49]. The microstructure consisted of the fcc phase; neither the bcc phase nor carbides were revealed in the TEM images. 

Tensile tests at room temperature of the SD, BD, and TD specimens of the as-produced using SLM Fe_65_(CoNi)_25_Cr_9.5_C_0.5_ alloy (Figure 9a, Table 3) showed some dependence of the mechanical properties on the orientation. The BD specimens possessed both the lowest yield strength (YS = 510 MPa) and tensile elongation (TE = 32%). The ductility of the TD and SD samples was rather close (TE = 37–38%); however, the strength of the SD samples was higher (YS = 600 and 560 MPa for SD and TD, respectively).

Only TD specimens were tested at cryogenic temperatures (Figure 9b, Table 3) because of the relative similarity of the mechanical properties of the specimens of different directions. The YS value of the alloy at 77 K was noticeably higher than that at room temperature (680 vs. 560 MPa), while ductility remained quite good (TE ~27%). However, the ultimate tensile strength (UTS) of the alloy at 77 K reached ~1800 MPa, in contrast to 640 MPa at room temperature.

In addition, the curve obtained during tension at 77 K demonstrated quite a pronounced stage of yielding (till ε ≈ 10%) at a relatively constant stress of ~735 MPa, followed by intensive strengthening. This behavior differed noticeably from the typical behavior of polycrystals curves obtained at room temperature (Figure 9a,b).

Strain hardening rates were also found to be quite different depending on the temperature (Figure 9c). While at room temperature, strain hardening dropped to ~1.7–1.5 GPa and then became relatively stable (which also looks quite typical of metallic polycrystals), at 77 K, the curve had two domains with a considerable increase in value of strain hardening (to ~6–10 GPa) on the second domain. The intensive formation of the bcc phase during deformation and the “dropped-then-recovered” behavior of the dσ/dε curve were most likely associated with the activation of the TRIP effect [37,50,51].

Microstructure analysis also showed distinct differences in the structure of the as-produced Fe_65_(CoNi)_25_Cr_9.5_C_0.5_ alloy specimens tested at room and cryogenic temperatures (Figure 10). EBSD maps after tensile testing at room temperature revealed a noticeable increase in the fraction of the bcc martensite phase (to ~14%) (Figure 10b). Some elongation of the bcc grains and twin formation in the fcc phase can also be observed (Figure 10a). A decrease in the deformation temperature to 77 K led to a much more intensive martensite formation, so that the fraction of the bcc phase after the tensile test was found to be ~91% (Figure 10d). The surviving fcc phase formed isolated conglomerates. The formation of martensitic laths resulted in a considerable microstructure refinement, with the average martensitic laths thickness increasing from 1 μm (after tensile testing at 293 K) to 1.5 μm (after tensile testing at 77 K) (Figure 10b,d).

To shed light on the unusual deformation behavior of the Fe_65_(CoNi)_25_Cr_9.5_C_0.5_ alloy during deformation at 77 K (Figure 9b), interrupted tensile tests at cryogenic temperatures were additionally performed; EBSD microstructure analysis was carried out at ε = 5% and ε = 10% (Figure 11). At ε = 5%, the martensite was found to be formed along the fcc grain boundaries (as agglomerates of relatively small laths) and in the grain interiors in the form of long lamellae declined ~45° from the tension axis (Figure 11b). The volume fraction of the bcc martensite was found to be 22 %. An increase in the strain to 10% leads to an increase in the volume fraction of the martensite to 32% (Figure 11d). At this stage of deformation, the bcc phase also precipitated both along the grain boundaries and within the grains. After fracture, the martensite phase comprised 92%. Note that the results obtained (Appendix A) suggest chemical uniformity between the bcc and fcc phases in accordance with the previous literature on similar alloys [7].

Charpy tests were also carried out both at room temperature and at 77 K. The Fe_65_(CoNi)_25_Cr_9.5_C_0.5_ alloy in the as-printed condition demonstrated fracture toughness values of 660 kJ/m^2^ and 430 kJ/m^2^ at room and cryogenic temperature, respectively (Table 4). The dimple fracture surfaces of all specimens (see Appendix A) after the impact tests and rather a high level of the fracture toughness suggested ductile failure at both temperatures.

## 4. Discussion

The use of additive technologies in industry promises numerous advantages, which are mainly associated with the possibility of producing various complex-shaped parts almost without machining [43,49,52,53]. At the same time, the performance of the products obtained using additive manufacturing (including SLM) depends substantially on the parameters of the process. In ideal cases, it is possible to obtain metallic parts with almost zero porosity and with mechanical properties which would be close to those of counterparts obtained by more convenient methods (e.g., casting, thermomechanical treatment, and machining). Although there are many parameters which can be varied during SLM [54,55,56], some of them (laser beam power, beam velocity, overlapping, layer thickness, and defocusing) can be combined into energy density imparted to the metallic powder [45,46,57]. In the present work, only two parameters were varied (beam power and beam velocity), while the others were fixed. Expectedly, both too high and too low energy densities did not result in a proper quality of the printed specimens (Figure 3). The most appropriate values of energy density to produce specimens with porosities less than ~0.5% were found to fall in the interval of *E* = 75–97 J/mm^3^ (Figure 4, Table 2). The obtained results agree with those reported in [46], where the range of optimal E was found to be from 62.5 to 115.6 J/mm^3^. However, even at the same value of *E* = 83 J/mm^3,^ the porosity can vary noticeably (Figure 4) depending on the involved parameters. Based on Figure 4, the optimal mode, due to which the minimum volume fraction of pores is achieved, is—*P* = 200 W, *V* = 1600 mm/s (*E* = 83 J/mm^3^). It was found that an increase in scanning speed (up to 1800–2200 mm/s) at low powers (100–150 W) resulted in an increase in the volume fraction of pores due to insufficient energy for complete fusion. At the same time, low scanning speeds (from 400 to 1400 mm/s) and high power (300 W) led to splashing of the remelted powder particles, overburning, and remelting of the substrate (Figure 3), which also contributed to a pronounced increase in the porosity of the alloy. The effect of various parameters on the structure and properties of metallic materials obtained by SLM has been comprehensively discussed elsewhere [47,54,58,59]. In the framework of the present study the main aim of the parameters selecting was primarily associated with obtaining the lowest porosity.

The microstructure of the produced (mode: *P* = 200 W and *V* = 1600 mm/s) sample is characterized by cells (Figure 8). The cell structure formation during SLM is usually ascribed to thermal distortions during SLM, which are caused by the constraints surrounding the melt bath and thermal cycling [60,61]. The dislocation (cell) structure forms within the grains after solidification. The specimens produced using the optimal parameters show at room temperature (for the TD direction) quite attractive mechanical properties: YS = 560 MPa, UTS= 640 MPa, TE = 37%, and fracture toughness of 660 kJ/m^2^ (Table 3 and Table 4). However, due to the presence of some metallographic texture in the specimens obtained by SLM, the strength can vary (±50 MPa) depending on the direction. Similar results were previously reported for various alloys produced by SLM [62,63].

The mechanical behavior of the Fe_65_(CoNi)_25_Cr_9.5_C_0.5_ alloy obtained by SLM was found to be quite different from that produced by casting (the structure and mechanical properties of the as-cast Fe_65_(CoNi)_25_Cr_9.5_C_0.5_ alloy were discussed in our earlier work [7]). At a comparable level of porosity (~0.1% in both cases, Figure 4 and [7]), the SLM alloy at room temperature showed considerably higher yield strength (510–600 MPa vs. 155 MPa) and lower ductility (32–37% vs. 65%). Besides, the SLM alloy possessed lower strain hardening (in contrast to that of the cast alloy) due to which the values of UTS of both compositions of the alloy were quite similar to each other(~600–700 MPa) (Table 3). 

Sometimes, the higher strength and lower ductility of the materials produced using additive manufacturing are ascribed to a higher level of impurities in comparison with that in as-cast alloys [64,65]. Indeed, examination of the chemical composition suggests a 2–4 times increased level of interstitials (oxygen and nitrogen) in the specimen obtained by SLM (Table 1 and [7]). However, the contribution of these impurities in strength is not expected to be high [66,67]. Another reason for the higher strength can be associated with the developed substructure formation after rapid solidification during SLM (Figure 8). The increased dislocation density and cell structure contributed to the strength, thereby leading to increased yield stress and lower strain hardening, while the ultimate tensile strength almost did not change. Furthermore, the fine structure can also increase the stability of the fcc phase, thereby preventing martensite formation during cooling from melting [68,69,70,71]. As a result, less than 1% of the bcc phase was observed in the as-produces by SLM alloy (in comparison with ~35% in the as-cast alloy) (Figure 7b). Deformation of the alloy was accompanied by the fcc-to-bcc martensite transformation. However, the amount of the martensite was rather low (~14%, see Table 5) after straining at room temperature to fracture due to which the influence of this phase transition on the mechanical properties (strain hardening) was rather unpronounced. It is worth noting that a similar amount of the bcc phase and a modest TRIP effect were observed earlier during deformation at room temperature of the as-cast Fe_65_(CoNi)_25_Cr_9.5_C_0.5_ alloy [7].

A decrease in deformation temperature to 77 K also does not result in the formation of the bcc phase in the alloy (Table 5), thereby confirming the above suggestion about increased stability of the refined austenite. However, tension at this temperature leads to a gradual arising of the martensite and corresponding strengthening of the alloy (Figure 10 and Figure 11 and Table 5). Data shown in Figure 12 suggest a nearly linear dependence of the flow strength of the alloy on the fraction of the bcc phase (for the point relevant ε≈20%, IPF and phase maps are presented in the Appendix A). Similar results were earlier observed hereafter [10,72,73,74,75]. 

The obtained results show that SLM of TRIP medium entropy alloys can be used for achieving a good combination of mechanical properties at room and particularly at cryogenic temperatures. However further efforts are needed to determine the most proper technological windows for SLM of a variety of Fe-rich medium entropy alloys with the TRIP/TWIP effect.

## 5. Conclusions

The structure and mechanical behavior at room and cryogenic temperatures of the Fe_65_(CoNi)_25_Cr_9.5_C_0.5_ medium-entropy alloy obtained by the selective laser melting (SLM) method was studied. The following conclusions were drawn:The optimal parameters of SLM (*P* = 200 W, *V* = 1600 mm/s) of the Fe_65_(CoNi)_25_Cr_9.5_C_0.5_ medium-entropy alloy resulted in residual porosity of ~0.5%.Despite the similar microstructure of the alloy, mechanical properties of the scanning direction, transverse direction, and building direction (SD, TD, and BD, respectively) specimens showed some anisotropy in mechanical properties (strength varied ± 50 MPa). The tensile properties at room temperature for the TD direction were: YS = 560 MPa, UTS = 640 MPa, TE = 37% (yield strength, ultimate tensile strength, and tensile elongation, respectively).A decrease in the temperature to 77 K resulted in a significant increase in both the yield strength and ultimate tensile strength to 680 MPa and 1800 MPa, respectively, and a subtle decrease in ductility (TE = 26%). The obtained high cryogenic properties are associated with the development of the transformation-induced plasticity (TRIP) effect due to the face-centered cubic to body-centered cubic (fcc-to-bcc). The microstructure of the alloy strained at 77 K became almost completely martensitic (the volume fraction of the martensitic phase reached 92%).

## Figures and Tables

**Figure 1 materials-16-03193-f001:**
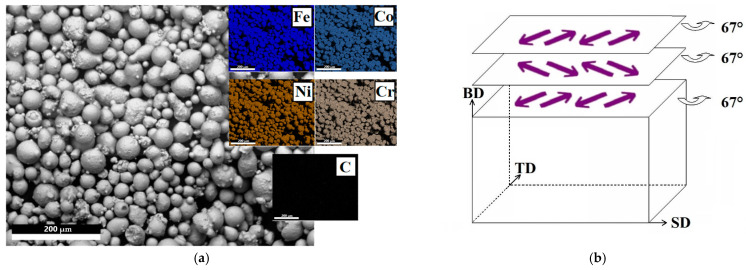
Morphology and EDS maps of Fe_65_(CoNi)_25_Cr_9.5_C_0.5_ powder (**a**) and schematic illustration of the scanning strategy (**b**). BD, SD, and TD indicate building direction, scanning direction, and transverse direction, respectively; it should be noted that each laser beam track was declined 67° from the macroscopic scanning direction.

**Figure 2 materials-16-03193-f002:**
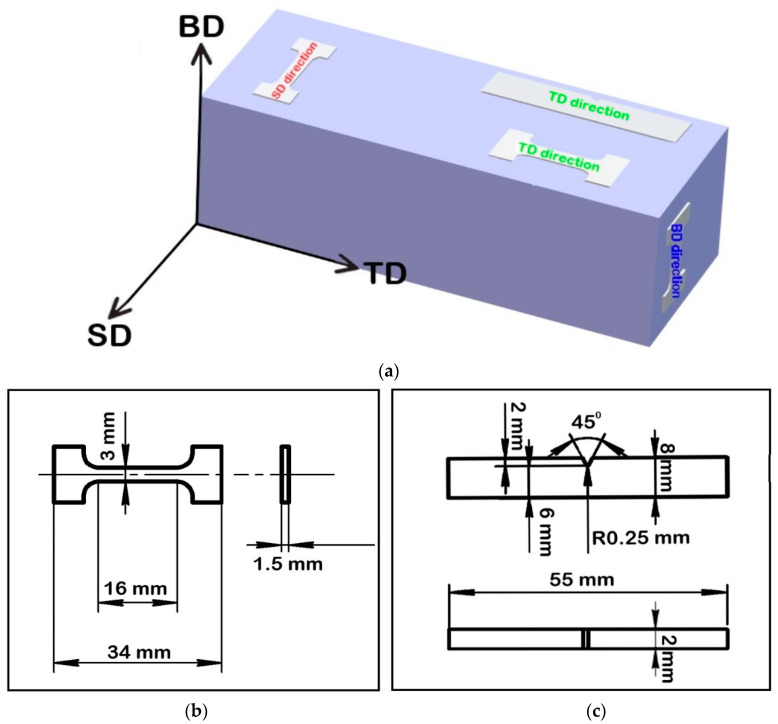
(**a**) Schematic of the geometry of the produced specimens. BD, SD, and TD indicate building direction, scanning direction, and transverse direction, respectively. The (**b**) tensile and (**c**) impact specimens (and their dimensions) are illustrated.

**Figure 3 materials-16-03193-f003:**
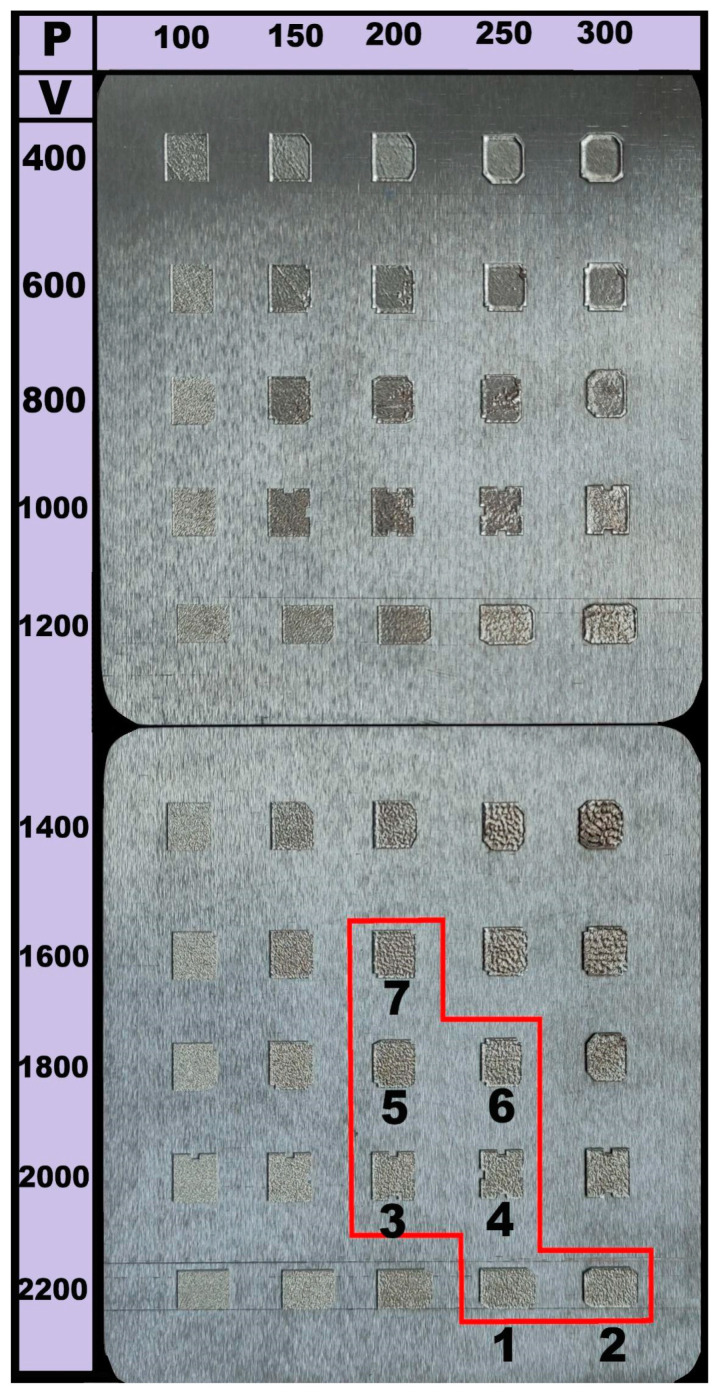
Selected parameters of SLM (laser beam power (*P*) and velocity (*V*)).

**Figure 4 materials-16-03193-f004:**
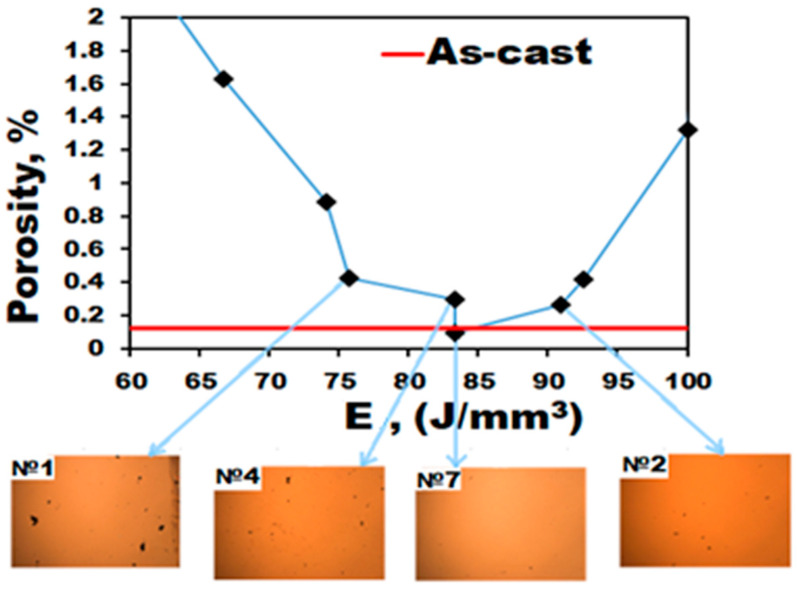
Dependence of the pore volume fraction on the energy density (*E*) and OM images of some representative specimens.

**Figure 5 materials-16-03193-f005:**
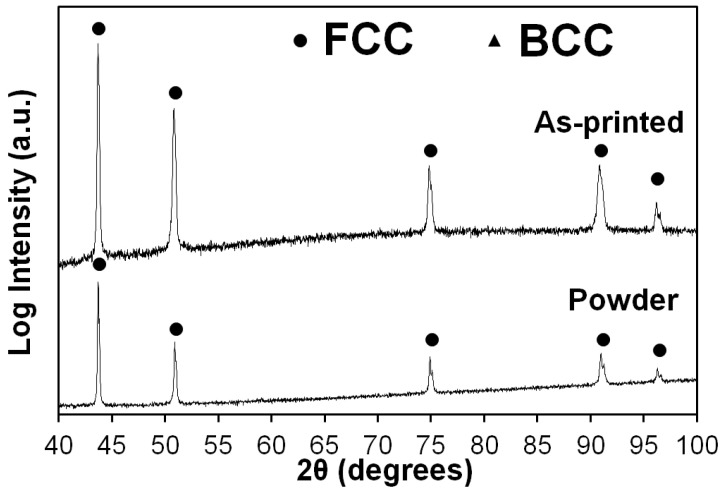
X-ray diffraction patterns of powder and as-printed Fe_65_(CoNi)_25_Cr_9.5_C_0.5_ alloy.

**Figure 6 materials-16-03193-f006:**
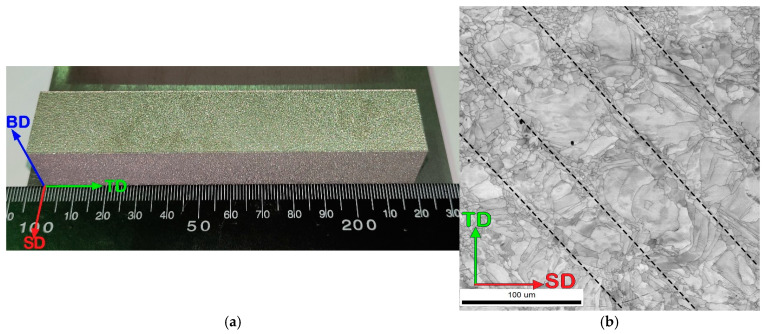
Photograph (**a**) and microstructure (EBSD image quality (IQ) map) (**b**) of the as-produced (mode: *P* = 200 W and *V* = 1600 mm/s) Fe_65_(CoNi)_25_Cr_9.5_C_0.5_ alloy specimen (the black dotted lines indicate the laser beam tracks).

**Figure 7 materials-16-03193-f007:**
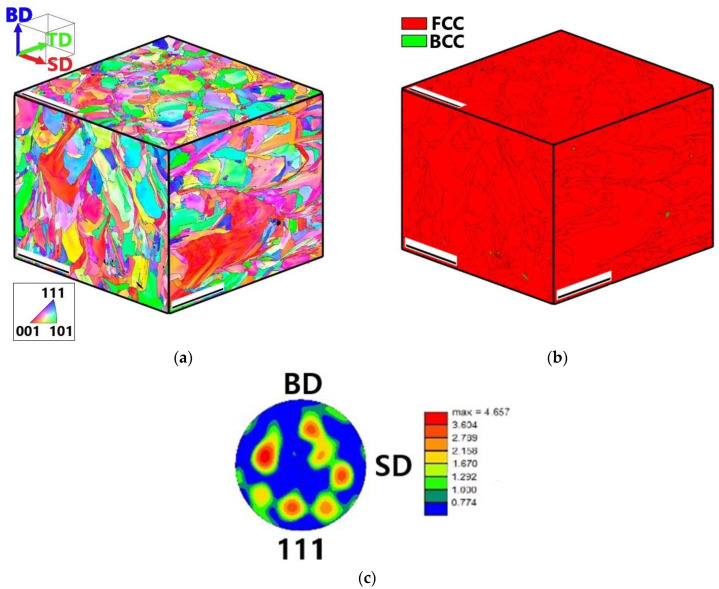
Quasi-3D microstructures of the as-produced (mode: *P* = 200 W and *V* = 1600 mm/s) Fe_65_(CoNi)_25_Cr_9.5_C_0.5_ alloy: IPF (**a**), phase maps (**b**) and (111) pole figure of the fcc matrix phase (**c**). The scale bars in (**a**,**b**) is 100 µm.

**Figure 8 materials-16-03193-f008:**
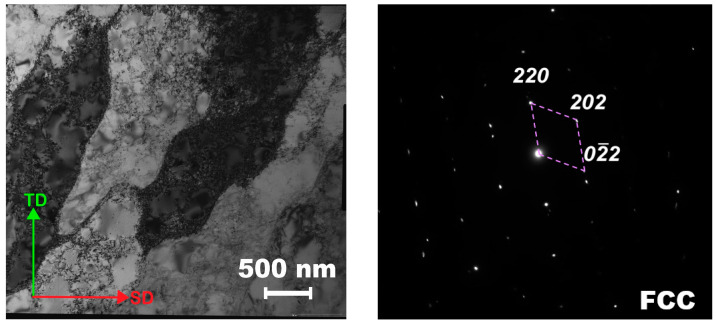
TEM image of the microstructure of the as-produced (mode: *P* = 200 W and *V* = 1600 mm/s) Fe_65_(CoNi)_25_Cr_9.5_C_0.5_ alloy.

**Figure 9 materials-16-03193-f009:**
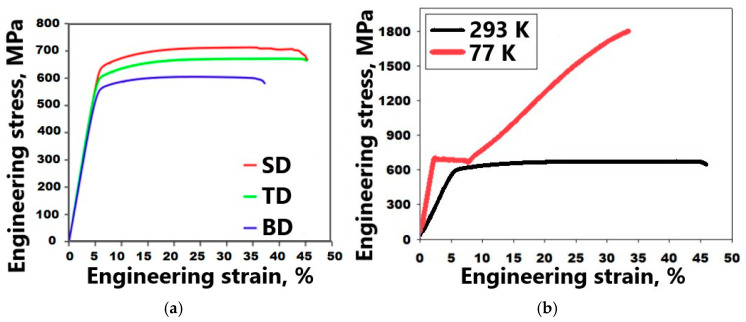
Stress-strain curves obtained during tensile tests at room temperature (**a**); comparison of the mechanical behavior of TD specimens at room and cryogenic temperature: stress-strain curves (**b**) and strain hardening (**c**) of the as-produced (mode: *P* = 200 W and *V* = 1600 mm/s) Fe_65_(CoNi)_25_Cr_9.5_C_0.5_ alloy.

**Figure 10 materials-16-03193-f010:**
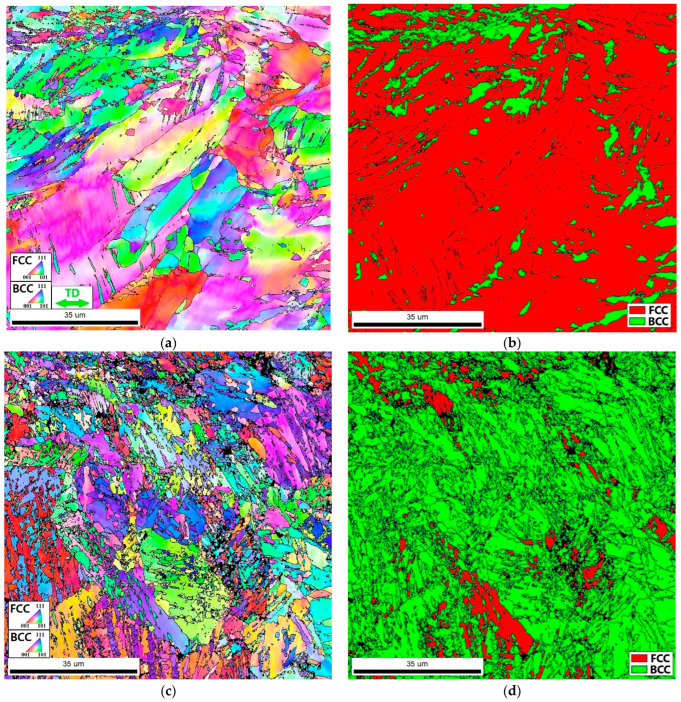
IPF (**a**,**c**) and phase (**b**,**d**) maps of the as-produced (mode: *P* = 200 W and *V* = 1600 mm/s) Fe_65_(CoNi)_25_Cr_9.5_C_0.5_ alloy after tension to fracture at room (**a**,**b**) and cryogenic (**c**,**d**) temperatures. The tension axis was horizontal in all cases.

**Figure 11 materials-16-03193-f011:**
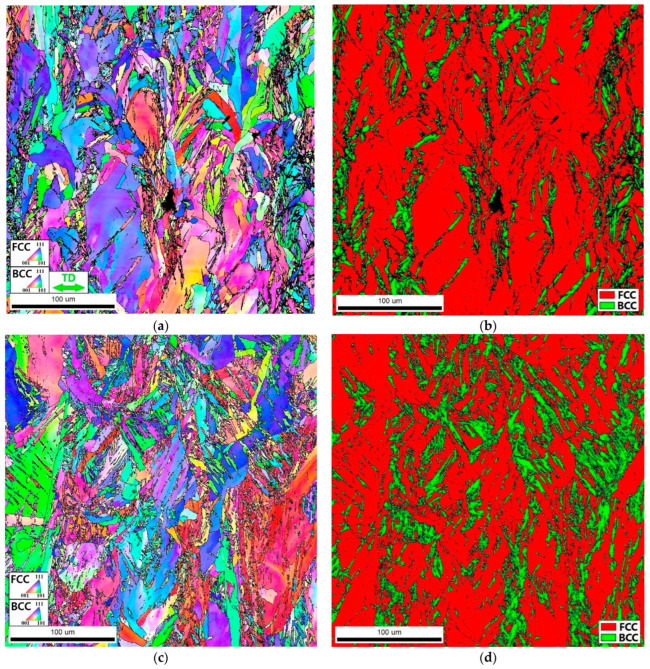
IPF (**a**,**c**) and phase (**b**,**d**) maps of the as-produced (mode: *P* = 200 W and *V* = 1600 mm/s) Fe_65_(CoNi)_25_Cr_9.5_C_0.5_ alloy after interrupted tension to ε = 5% (**a**,**b**) and ε = 10% (**c**,**d**) at cryogenic temperature. The tension axis was horizontal in all cases.

**Figure 12 materials-16-03193-f012:**
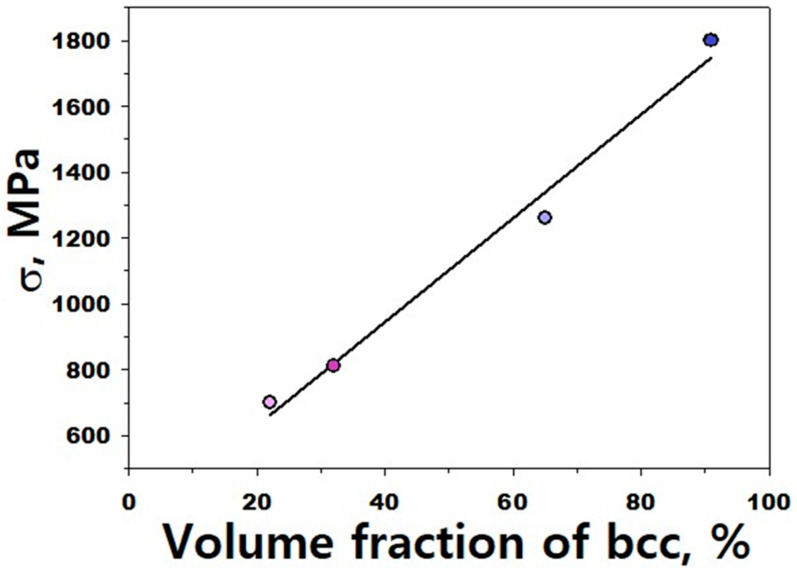
Strength of the Fe_65_(CoNi)_25_Cr_9.5_C_0.5_ alloy as a function of the martensite fraction.

**Table 1 materials-16-03193-t001:** Chemical composition of the as-produced alloy.

Composition	Fe	Co	Ni	Cr	C	O	N
Nominal (at.%)	65	12.5	12.5	9.5	0.5	-	-
Actual (at.%)	64.5	12.1	11.6	11.3	0.52	0.017	0.031
Actual (wt.%)	65.4	12.6	12.4	9.48	0.12	0.009	0.014

**Table 2 materials-16-03193-t002:** Selected parameters of SLM (laser beam power (*P*), velocity (*V*), and energy density (*E*) calculated per Equation (1)).

	#1	#2	#3	#4	#5	#6	#7
*P*, W	250	300	200	250	200	250	200
*V*, mm/s	2200	2200	2000	2000	1800	1800	1600
*E*, J/mm^3^	76 ± 1.5	90 ± 1.2	67 ± 0.6	83 ± 0.9	74 ± 1.1	93 ± 0.4	83 ± 0.5

**Table 3 materials-16-03193-t003:** Mechanical properties of the as-produced Fe_65_(CoNi)_25_Cr_9.5_C_0.5_ alloy.

	YS, MPa	UTS, MPa	TE, %
Sample TD, 293 K	560 ± 19	640 ± 11	37 ± 3
Sample TD, 77 K	680 ± 17	1800 ± 13	26 ± 1
Sample SD, 293 K	600 ± 13	710 ± 18	38 ± 2
Sample BD, 293 K	510 ± 16	600 ± 10	32 ± 2

**Table 4 materials-16-03193-t004:** Charpy V-notch impact energy and fracture toughness of the as-printed Fe_65_(CoNi)_25_Cr_9.5_C_0.5_ alloy obtained at room and cryogenic temperatures.

Testing Temperature, K	Fracture Toughness, kJ/m^2^	Charpy V-Notch Impact Energy, J
293	660 ± 16	10 ± 2
77	430 ± 12	7 ± 1

**Table 5 materials-16-03193-t005:** Volume fraction of the bcc phase depending on the condition of the alloy.

Temperature, K	Condition			
	As-Produced	Strained to ε ≈ 5%	Strained to ε ≈ 10%	Strained to Fracture (ε ≈ 26%)
293	0.5%	-	-	14%
77	0%	22%	32%	91%

## Data Availability

The data presented in this study are available on request from the corresponding author. The data are not publicly available because it is a part of an ongoing study.

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
