# Peer review of "Mechanical Behavior of a Medium-Entropy Fe65(CoNi)25Cr9.5C0.5 Alloy Produced by Selective Laser Melting"

_materials, 2023, doi:10.3390/ma16083193_

Round 1
Reviewer 1 Report
The authors investigated the mechanical behaviour of Fe65(CoNi)25Cr9.5C0.5 MEA produced by the SLM process when subjected to tension at room and cryogenic temperatures. This is good work considering the fact that limited studies have been carried out in this area. However, the manuscript can be further improved by addressing the followings:
- Please, ensure that all abbreviations well defined starting from the abstract down to the conclusion section
- Kindly specify the flow rate of the argon gas utilized as shielding gas
- Full dimensions of the tensile sample should be provided in Fig.2
- The role of building parameters (power and beam scan speed) on porosity still needs to be clearly elucidated. The porosity results have been presented but the scientific reasons behind this occurrence are missing
- The captions of Figs.6, 7, 8, 9, 10, 11 need to show/reveal the process parameters employed for the fabrication of the samples
Reviewer 2 Report
Paper 'Mechanical behavior of a medium entropy Fe65(CoNi)25Cr9.5C0.5 alloy produced by selective laser melting' is aiming to study the porosity and microstructure of L-PBFed Fe65(CoNi)25Cr9.5C0.5 alloy and their influence on mechanical behavior. The porosity and microstructure were investigated by OM, SEM and TEM as well as EBSD. Although the writing was good, some questions should be added:
1. In Fig.7 and Fig.8, the authors should give the element distribution and verify the element composition of FCC and BCC phase.
2. From TEM and EBSD images in Fig.8 and Fig.10, the sub-microstructure was clearly found. The authors should explain the formation mechanism of sub-microstructure, and its relationship with grain growth.
3. In Fig.13, the strength of Fe65(CoNi)25Cr9.5C0.5 alloy as a function of the martensite fraction. Whether the martensite fraction can be adjusted by changing the processing parameters?
